# Runoff Mitigation in Croplands: Evaluating the Benefits of Straw Mulching and Polyacrylamide Techniques

Eli Argaman [1,*] and Ilan Stavi [2]

1   Soil Erosion Research Station, Soil Conservation and Drainage Division, Ministry of Agriculture & Rural Development, Bed Dagan 50250, Israel
2   Dead Sea and Arava Science Center, Yotvata 88820, Israel; istavi@adssc.org
*   Correspondence: eliar@moag.gov.il

**Abstract:** Water loss through surface runoff is a significant constraint for rainfed agricultural lands across the Mediterranean region. Using straw-mulch cover (SMC) as a runoff mitigator has been successfully utilized to negate the impact of raindrop splashing. However, this practice is uncommon due to the high demand for crop residue as feed or fodder for livestock. Therefore, the application of synthetic polyacrylamide (PAM) has become a common practice. Although many studies have shown the positive impact of PAM on runoff control, most were conducted under laboratory conditions, where interactions with crop phenology and runoff dynamics were disregarded. In this study, on-site rainfall simulation was used to determine the efficiency of PAM and SMC to control runoff from foxtail millet (*Setaria italica*) fields under three seasonal conditions: (1) high-intensity rainfall, characteristic of autumn, on bare soil surfaces; (2) moderate-intensity rainfall, characteristic of winter, following crop tillering; and (3) high-intensity rainfall, characteristic of spring, following the flowering phase. The effect of SMC during the autumn and spring simulations was significantly better than that of the PAM and control treatments. For the winter simulation, runoff rates and runoff ratios were similar for all treatments. The most prominent finding was obtained for the spring simulation, where SMC yielded no runoff, whereas the PAM and control treatments yielded similar runoff rates and runoff ratios.

**Keywords:** conventional tillage; polyacrylamide; rainfall simulator; runoff; straw-mulch cover

## 1. Introduction

Intensive rainfed agriculture is the primary land-use over large areas in the Mediterranean regions, while irrigated agricultural lands are a small fraction of the total area. The total area of the Mediterranean countries extends over 877 million ha of land, of which approximately 28% is used for agricultural production, mainly cereals (e.g., wheat, barley, and maize). Land degradation processes result from anthropogenic factors adversely affect the functioning of agricultural systems. These processes may be gradual and often undetectable in the short term [1]. Land degradation processes include decreased soil organic matter, reduced aggregate stability, loss of vegetation cover, and accelerated water loss from agricultural fields through over land flow. Loss of water through runoff is indeed a major constraint that poses a multifaceted and significant challenge across the diverse landscapes of the Mediterranean region, which is characterized by dry summers and cool, wet winters [2]. Throughout the region, limited, sporadic rainfalls and high evaporation rates pose a significant challenge to agricultural productivity [3], primarily because sealed surfaces and impermeable soil layers impair water infiltration [2]. This process triggers soil erosion, particularly on bare soils during the early growing season, and reduces soil fertility and agricultural productivity [4,5]. Among the consequences of runoff are the loss of valuable nutrients, depletion of soil horizons, and reduction in crop yield productivity [6,7]. Specifically, the Mediterranean region is vulnerable to climate change impacts,

under which excessive runoff may intensify land degradation processes, reducing the resilience of agricultural systems [8,9].

In rainfed agricultural lands across the Mediterranean region, where high-intensity rainstorms are frequent during autumn and spring [10,11], soil-conservation measures (e.g., mulching, cover cropping, and conservation tillage) can significantly mitigate the loss of runoff water and improve crops' water-use efficiency, simultaneously reinforcing nutrient cycling and increasing aggregate stability [12,13]. Further, long-term climatic trends across the eastern Mediterranean region indicate that precipitation variability is increasing, resulting in extensive drought years, which impact both the soil-water availability and runoff-rainfall ratio [1].

As in other Mediterranean regions, many of the rainfed clayey agricultural lands of the northern Israeli valleys are on moderate slopes (5.0–10.0%) that experience severe soil erosional processes following heavy rainstorms. Runoff generation and soil erosion in these croplands are frequent and accelerated by conventional tillage (CT) and even by reduced-tillage (RT) systems. In CT systems, the upper soil profile (20 to 30 cm) is altered by moldboard plowing that breaks the soil clods, inverts the soil horizons, buries crop residues, and eliminates weeds [14,15]. Harrowing (disking) is another practice, with the main difference being a negative angle of penetration into the soil (as opposed to the positive angle with the moldboard plow), which carries the risk of accelerated soil grinding. Intensive tillage increases runoff and soil erosion [16,17], reduces soil-water content in the rhizosphere, and eliminates the soil's A horizon [18,19]. During the intense rainstorms characteristic of the Mediterranean region, raindrop impact may cause clay dispersion [20–23], forming physical soil crusts that decrease infiltration rate and increase runoff generation and soil erosion [24–27].

Among the common soil-conservation practices (e.g., cover crops, strip cropping, mulching, and no-till farming), straw-mulch cover (SMC) has been successfully utilized to negate the impact of raindrop splash energy, reducing soil crusting and sealing in intensive agricultural fields [28–31], mitigating soil-water evaporation [32,33], improving infiltration capacity [34], regulating soil temperature, and improving water retention [31,35,36]. An additional benefit of the SMC practice is its positive impact on the soil organic matter pool (SOM) [37,38]. SOM accelerates the soil's macro-aggregation and structural formation by binding soil particles [36], decreasing the ground surface's susceptibility to runoff generation and soil erosion [39]. Although the effectiveness of mulch covers in reducing runoff is well known [40–43], this practice is uncommon in many parts of the world, mainly due to high demands for crop residue as feed or fodder for livestock, which reduces its availability due to its price. Further, some studies revealed that the application of SMC in laboratory rainfall simulations does not reduce runoff generation compared to bare soil [44], insignificantly reduces runoff discharge [45], and might accelerate soil loss under high-intensity rainfall conditions [46].

Polyacrylamide (PAM) is among the most common synthetic soil conditioners that can stabilize the soil surface by clumping fine soil particles, thus reducing irrigation-induced soil erosion [47,48]. Many studies have shown that PAM decreases runoff and erosion rates while increasing infiltration and improving water retention [49–51]. Other studies have reported the high efficiency of PAM in hindering crust formation by raindrop impact [52,53], stabilizing soil aggregates [54–56], improving nutrient retention and suppressing weed growth [57], and increasing the shear stress of the soil surface [58]. At the same time, under certain conditions, some limitations were reported. For example, Malik and Letey [59] reported that PAM's sorption capacity limits its ability to diffuse into montmorillonitic clayey soil aggregates. Lu et al. [60] found that increased SOM content in clayey soils reduces PAM sorption capacity due to increased electrostatic repulsion between the PAM molecules and soil particles. Moreover, Jia et al. [61] reported that, compared to other soil stabilizers, PAM was less effective in inhibiting crust formation. A laboratory rainfall-simulation study by Shainberg et al. [62] on vertisols obtained from the northern Israeli valleys showed that applying 20 kg ha$^{-1}$ PAM increased the infiltration rate from 3.0 to

20.5 mm h$^{-1}$. At the same time, Soupir et al. [63] reported that applying dry (granular) and wet PAM at 20 kg ha$^{-1}$ on clayey soils increased the runoff rate. It seems that the efficiency of PAM in regulating water runoff is site-specific and depends on the combination of soil properties and cropland management, as well as crop on hybrids and phenology. Further, the costs of PAM might negatively affect its application over large cropland fields, particularly under Mediterranean climate, where intensive rainstorms may occur during the autumn while most agricultural fields are bare and exposed to raindrop impact.

The objective of this controlled simulated rainfall field study was, therefore, to assess the impact of SMC (at a 400 kg ha$^{-1}$ application rate) and PAM (at a 20 kg ha$^{-1}$ rate) on runoff generation, runoff rate (RO), and runoff-rainfall ratio (RR) along a growing season for cereals grown in clayey soil in Harod Valley, Israel. The valley's climate poses challenges related to water scarcity and limited annual rainfall, requiring adequate water management and adopting soil conservation practices. The study hypothesis was that the efficiency of SMC in controlling runoff generation and reducing surface water loss throughout the growing season would be greater than that of PAM.

## 2. Materials and Methods

### 2.1. Study Site Description

The study was conducted in Harod Valley (32°33′ N, 35°22′ E, 20 m.a.s.l.), extending over 40 km$^2$ (Figure 1). The valley is characterized by a Mediterranean climate, featuring hot and dry summers and moderate, relatively wet winters. The average annual rainfall (for the hydrological year, starting in September and ending in the following April) is 405 ± 124.6 mm [64]. The annual average number of rainy days (>1 mm) is 51 ± 9.5. During autumn and spring, mean rainfall values are similar (63.8 ± 7.61 and 66.5 ± 5.77 mm, respectively, $p$ = 0.984), whereas, during the winter, they are significantly higher (273.4 ± 17.27 mm, $p$ < 0.0001). High-intensity rainstorms commonly occur in the autumn and spring [65]. The probability of rainfall intensity exceeding 30.0 mm/h (per 30 min) in autumn and spring is 10% (once in 10 years). During the winter, intensities are significantly lower, with a 1–2% probability of high-intensity rainfall (once every 50–100 years). Early-season (autumn) rainstorms occur mainly when the soil surface is bare, resulting in high erosion rates. Topography comprises moderately inclined plains, ranging from 2–6%. The soil parent materials are calcareous sediment and basalt, with an average profile depth of ~40 cm. The soil texture is clayey (>40% clay content); its properties are detailed in Table 1. According to the USDA soil taxonomy, the soil is classified as Vertisol, clay-rich soils that shrink and swell with moisture content changes that form deep, wide cracks [66]. The dominant land use is field crops, either rainfed or irrigated.

### 2.2. Plot and Treatment Setup

To determine the effects of granular anionic PAM with high molecular weight (SupoerFloc®—Superfloc 110-c Series A-100, 'Hadar'—https://www.haddar.co.il/, accessed on 20 June 2023) and SMC at a rate of 20.0 and 400.0 kg ha$^{-1}$, respectively, on runoff generation and rate, we constructed nine runoff plots (three replicate plots per treatment), with representative dominant slopes of 6.15 ± 1.06%. Each plot measured 1.44 m$^2$ (1.20 m × 1.20 m). All plots faced south. The experiments were performed with a portable rotating-disk rainfall simulator [67] at the end of the summer season (between mid-July and late August). The simulator was suspended 2 m above the surface, wetting a circular area of 2.10 m$^2$. The median raindrop size was 1.9 mm, and its terminal falling velocity was 6.02 m s$^{-1}$. Before the experiment, the field was chiseled to 10 cm depth to achieve a uniform starting point for all treatment plots. The plots were seeded with foxtail millet (*Setaria italica* (L.) P. Beauvois), a common crop across the region, and PAM and SMC were applied according to the experimental design (Figure 2). The surface cover of the SMC plot was 35.0%, while the PAM and control plots remained bare. The rainfall (using water of EC 0.0 ds m$^{-1}$) simulated three seasonal and phenological phases: (1) autumn—high-intensity rainstorm simulation, at a rate of 35 mm h$^{-1}$, and at a kinetic energy of 21.95 J mm$^{-1}$ m$^{-2}$,

under bare surface conditions, prior to plant emergence; (2) winter—moderate rainstorm intensity, at a rate of 14 mm h$^{-1}$, and a kinetic energy of 17.84 J mm$^{-1}$ m$^{-2}$, following tillering; and (3) spring—high-intensity rainstorm simulation, similar to that of the autumn session, following the flowering phase. All plots were identically irrigated with salt-free water (EC 0.0 ds m$^{-1}$) by sprinklers between the simulations every three days to achieve field capacity and avoid plant-water stress. The distribution of Christiansen's Uniformity coefficient (CU; based on 36 rain gauges in and around the simulator's wetted area), a measure of the uniformity of rainfall distribution of the rainfall simulator system, was monitored three times, over intervals of 30 min each, for each of the simulations. The CU results revealed a CU of ~84% for the 35 mm h$^{-1}$ rainstorm and ~86% for the 14 mm h$^{-1}$ rainstorm.

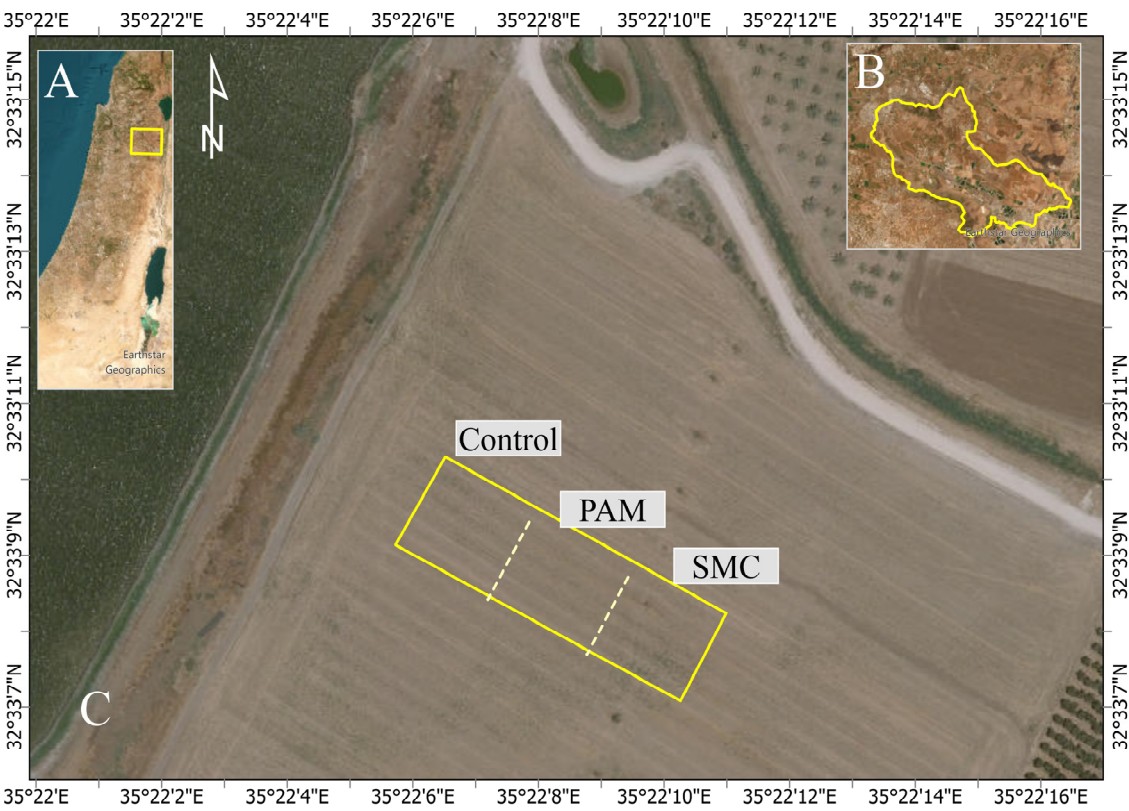

**Figure 1.** A—regional map of the watershed location; B—the Harod watershed boundaries; C—demarcation of Control, PAM, and SMC plots for the controlled rainfall simulation.

**Table 1.** Soil physical and chemical properties.

| Texture | Clay (%) | SOM (g kg$^{-1}$) | pH | EC (ds m$^{-1}$) | SAR | CaCO$_3$ (%) | Na$^+$ (meq L$^{-1}$) |
|---|---|---|---|---|---|---|---|
| Clayey | 43.7 (1.87) | 15.8 (1.30) | 7.63 (0.32) | 0.83 (0.39) | 0.55 (0.16) | 12.08 (1.38) | 1.13 (0.56) |

Abbreviations: SOM—soil organic matter; EC—electrical conductivity; SAR—soil adsorption ratio; CaCO$_3$—calcium carbonate content; Na$^+$—sodium content. Note: SE values are shown in parentheses.

For all experimental treatments, rainfall simulation lasted until a constant runoff rate was reached (for at least five consecutive measurements). The runoff from the plot outlet was pumped into a container at a frequency of 3–5 min, starting with the emergence of runoff until five constant replicates were obtained. Aboveground vegetation height, cover percentage, and volume were measured manually and photographed before each phase to assess their effect on runoff characteristics. The surface-cover percentage was assessed by supervised classification using ENVI 5.6 (L3Harris Geospatial, https://www.nv5geospatialsoftware.com/, accessed on 20 June 2023), based on a pixel-based maximum-likelihood algorithm. This simple and effective image-processing classifi-

cation method resulted in the highest overall accuracy (>90%) and minimal reliability of 0.86 (kappa coefficient), compared to other standard supervised classification algorithms (i.e., Mahalanobis Distance, Minimum Distance, Parallelepiped, and Spectral Information Distance). The classification's overall accuracy and kappa coefficient of agreement (varying from 0–1, where 0 represents no agreement and 1 presents perfect agreement) were visually validated using the generated orthophoto and against known field points used for accuracy analysis.

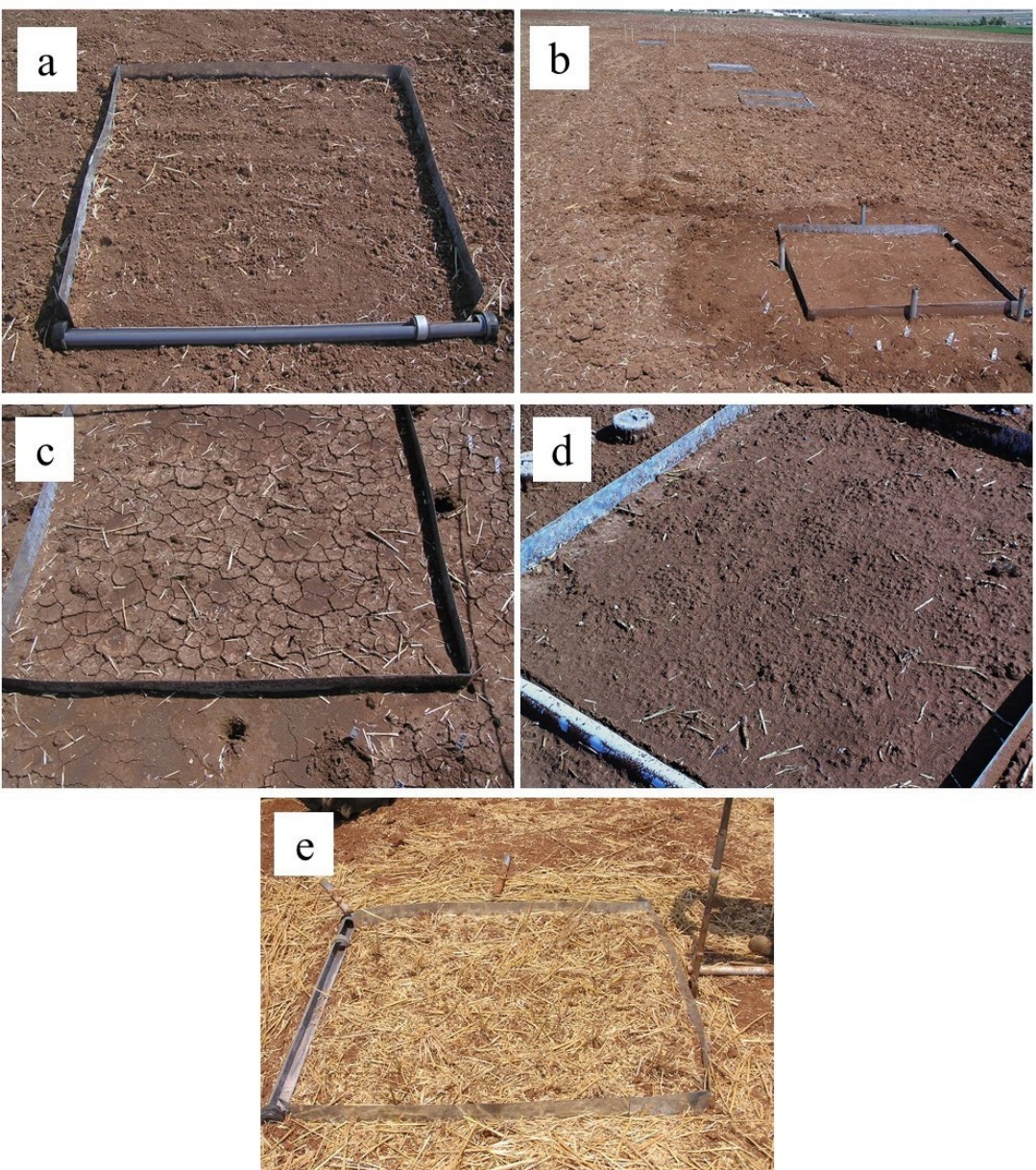

**Figure 2.** (**a**) Characteristic surface soil before treatment application. (**b**) Runoff plot after seeding and wetting. (**c**) Soil surface crusting of the control plot. (**d**) Surface view of the PAM application. (**e**) Surface view of the SMC application.

## 3. Results

A one-way ANOVA was used to assess the treatment (i.e., control, PAM, and SMC) effect on runoff throughout the cropping season.

### 3.1. Autumn Simulation

The simulated autumn rainfall over the bare surface of the control and PAM-treated soils and the SMC-covered soil produced notable variations in the measured parameters (Table 2). The average cumulative rainfall depth until runoff generation and the mean total rainfall depth to steady-state runoff for SMC were significantly ($p < 0.05$) higher than for the control and PAM treatments; the rain accumulation of the SMC treatment was 123.6 mm, compared to 50.38 and 45.02 mm for the Control and PAM, respectively. Rain accumulation values in the control and PAM treatments were similar ($p = 0.928$) (Figure 3). The lowest final RO was measured for the SMC treatment (24.44 mm h$^{-1}$, $p < 0.05$) and the highest for the PAM and control treatments (33.56 and 33.33, respectively). The mean RO was calculated as follows:

$$RO_{mean} = (\Delta(R_{acc} - RO_{gen}))/(simulation\ time)$$

where: $RO_{mean}$—mean runoff rate; $R_{acc}$—rainfall accumulated during the entire simulation; and $RO_{gen}$—cumulative rainfall depth at runoff generation.

**Table 2.** Summary table of runoff rate (RO), runoff ratio (RR), and vegetation results for autumn (AU), winter (WI), and spring (SP) rainfall simulations.

| Treatment | | RI (mm h$^{-1}$) | Time (min) | $R_{acc}$ (mm) | $RO_{gen}$ (mm) | $RO_{mean}$ (mm h$^{-1}$) | $RO_{final}$ (mm h$^{-1}$) | $RR_{mean}$ (mm mm$^{-1}$) | $RR_{final}$ (mm mm$^{-1}$) | VC (%) | Plant H (m) | Plant V (L m$^{-1}$) |
|---|---|---|---|---|---|---|---|---|---|---|---|---|
| AU | C | 35.0 | 85 (5.05) | 50.38 (3.85) | 22.66 (2.62) | 19.56 (1.16) | 33.33 (0.65) | 0.699 (0.22) | 0.933 (0.11) | — | — | — |
| | PAM | | 75.67 (2.26) | 45.02 (1.75) | 17.73 (0.96) | 21.60 (1.04) | 33.56 (0.98) | 0.739 (0.20) | 0.940 (0.17) | — | — | — |
| | SMC | | 210.33 (6.48) | 123.57 * (5.17) | 42.74 * (3.21) | 22.97 (1.30) | 24.44 * (2.36) | 0.392 * (0.30) | 0.685 * (0.39) | 35.0 (1.48) $^\dagger$ | — | — |
| WI | C | 14.0 | 288.0 (8.82) | 67.20 (4.26) | 35.87 (0.75) | 6.26 (1.40) | 4.20 (1.19) | 0.240 (0.17) | 0.300 (0.102) | 72.14 (1.01) | 0.35 (0.22) | 25.25 (1.90) |
| | PAM | | 210.0 (8.81) | 49.00 (4.25) | 18.67 (2.79) | 8.72 (1.09) | 8.68 (1.47) | 0.476 (0.38) | 0.620 (0.39) | 73.33 (1.17) | 0.31 (0.2) | 22.73 (1.53) |
| | SMC | | 240.67 (2.01) | 64.87 (3.99) | 21.82 (1.46) | 11.77 (2.03) | 3.46 (1.93) | 0.177 (0.199) | 0.247 (0.266) | 76.0 (2.47) | 0.205 * (0.17) | 15.58 * (1.92) |
| SP | C | 35.0 | 168.33 (8.65) | 100.2 (6.67) | 13.09 (2.07) | 29.86 (1.97) | 16.86 (1.33) | 0.307 (0.11) | 0.472 (0.22) | 81.4 (3.05) | 0.40 (0.22) | 32.56 (2.33) |
| | PAM | | 224.33 (9.96) | 133.48 (7.68) | 25.58 (2.62) | 27.91 (1.98) | 16.47 (2.28) | 0.278 (0.35) | 0.461 (0.38) | 77.16 (2.15) | 0.47 (0.21) | 36.26 (2.21) |
| | SMC | | 240.0 | 142.8 | — | — | 0.00 | 0.00 | 0.00 | 78.9 (2.60) | 0.525 (0.15) | 41.14 (1.38) |

Abbreviations: RI—rainfall intensity; Time—simulation time to initiation of steady-state runoff flow; Racc—rainfall accumulated during the entire simulation; ROgen—cumulative rainfall depth at runoff generation; ROmean—mean runoff rate; ROfinal—final runoff rate following steady-state runoff flow; RRmean—mean runoff ratio of ROgen to steady-state runoff; RRfinal—final runoff ratio; VC—vegetation cover; H—height; V—volume; C—control; PAM—polyacrylamide; SMC—soil mulch cover. Note: SE values are shown in parentheses. * Significant at $p < 0.05$. $^\dagger$ Corresponds to the percentage of mulch cover.

A significantly lower ($p < 0.05$) mean RO was observed for SMC compared to PAM and control treatments (Figure 3). SMC's mean and final RRs were significantly lower than those for the PAM and control treatments (Table 2, Figures 3 and 4).

### 3.2. Winter Simulation

The winter simulation was conducted following the tillering phase. All treatments' average surface vegetative coverage was similar ($p = 0.873$; Table 2). Plant height for the SMC treatment was significantly lower ($p < 0.05$) than for the control and PAM treatments. Similarly, plant volume—an indicator of the canopy's raindrop absorption capacity—was significantly lower ($p < 0.05$) for the SMC treatment than for the control and PAM treatments. The total cumulative rainfall, until reaching a steady-state RO, varied among treatments, being lowest for PAM and highest for the control, with a substantial root-mean-square deviation (RMSE) of 17.3 mm and a range of 18.2 mm ($p = 0.527$) between all replicates. In this scenario, the amount of rainfall that generated runoff was highest in the control treatment and lowest in the PAM treatment ($p = 0.06$). No significant differences ($p > 0.05$)

were observed between the treatments' mean and final RRs (Figures 3 and 4). The highest mean RR was measured for the PAM treatment and the lowest for the SMC treatment. A similar, albeit insignificant (*p* > 0.05), trend was measured for the final RR.

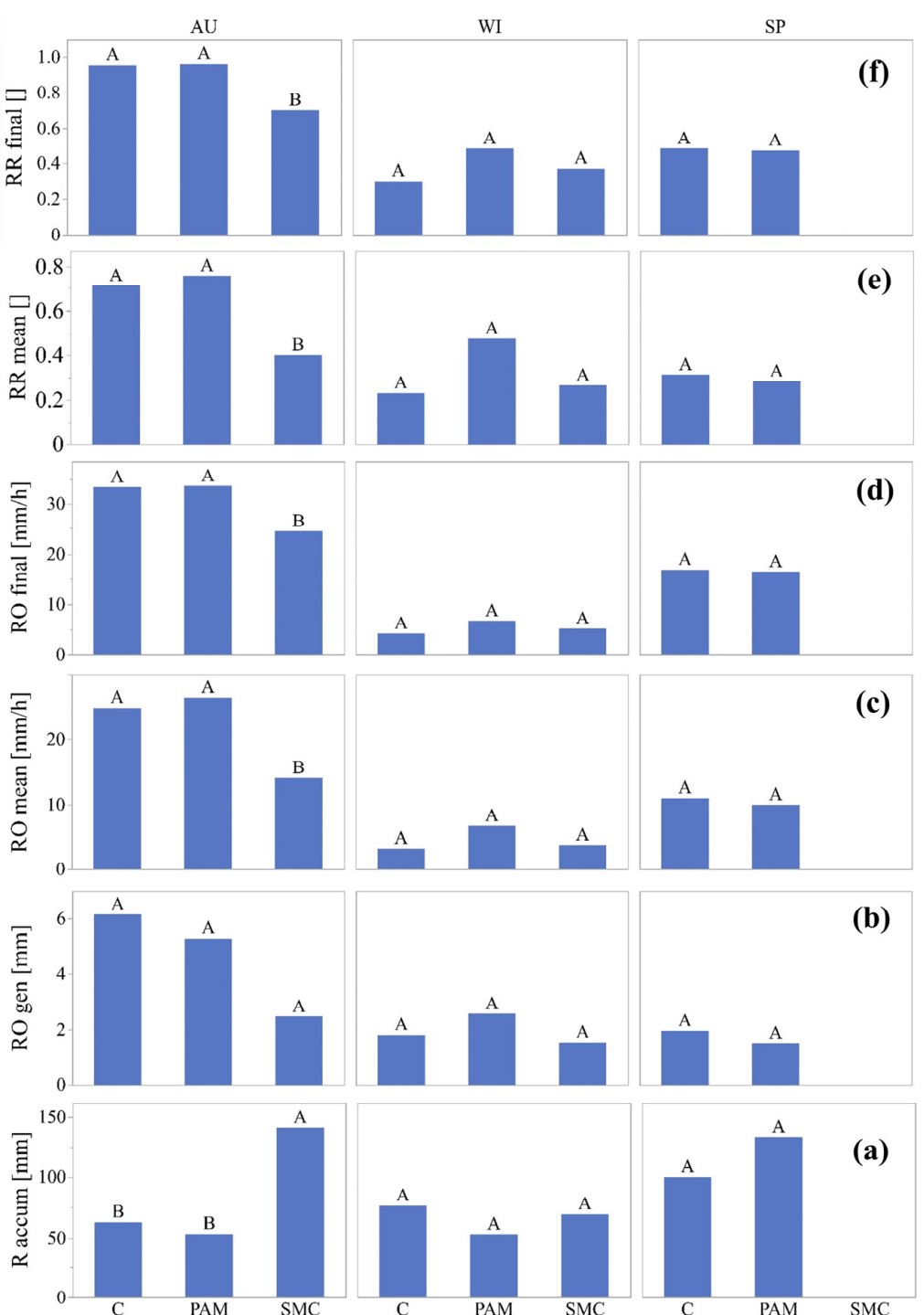

**Figure 3.** Results of the controlled experiment for the measured rainfall and runoff properties for each simulation phase (Autumn, Winter, and Spring). (**a**) mean runoff accumulation; (**b**) accumulative rainfall at runoff generation; (**c**) mean runoff rate during each simulation phase; (**d**) final runoff rate at each simulation phase; (**e**) mean runoff-rainfall [RR] ratio during each simulation phase; (**f**) final runoff-rainfall [RR]. Different letters indicate a significant difference between treatments.

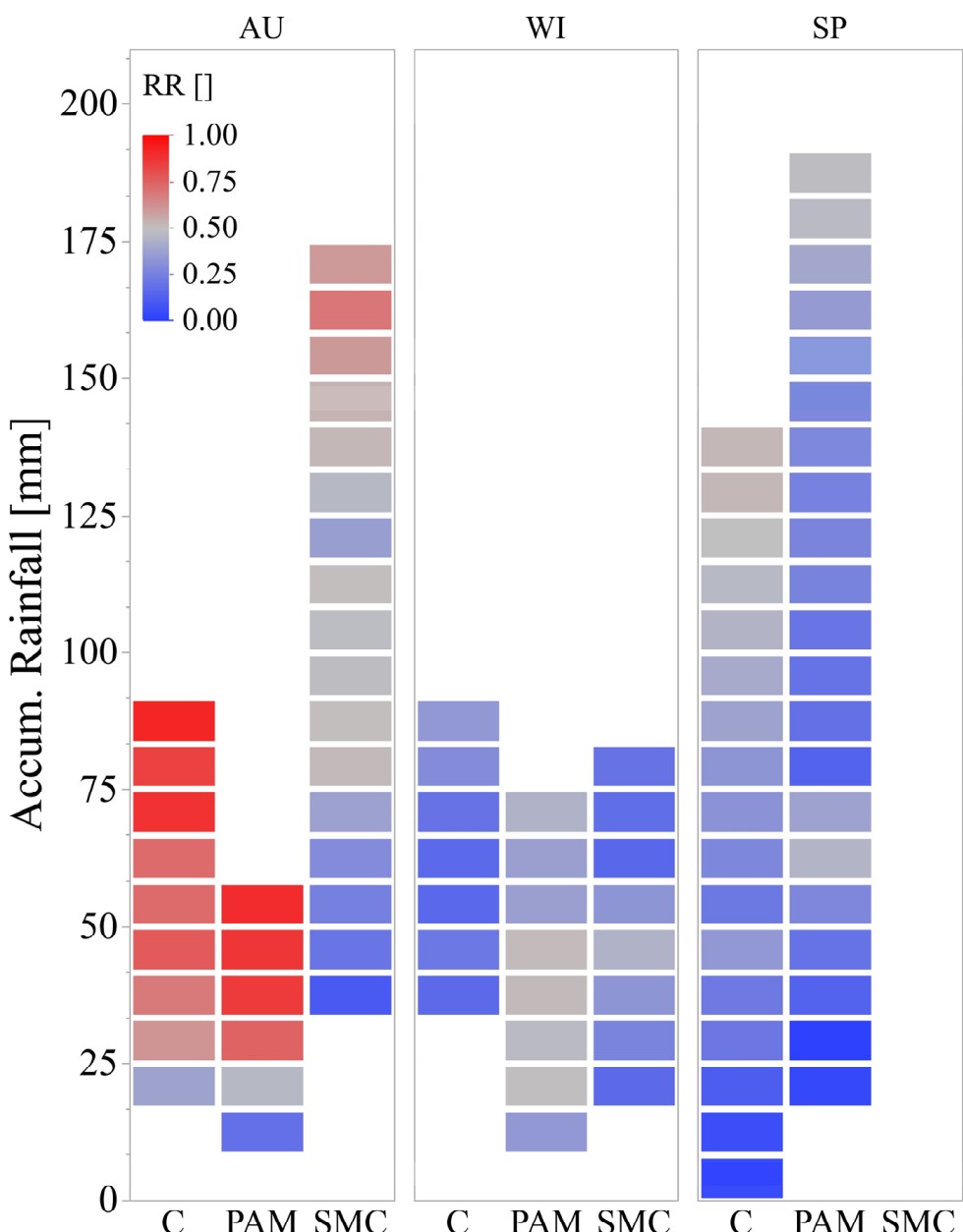

**Figure 4.** Rainfall-Runoff (RR) distribution during the Autumn (AU), Winter (WI), and spring (SP) simulations.

### 3.3. Spring Simulation

The spring rainstorm simulation was conducted following the flowering phase. Mean vegetation cover varied insignificantly among the treatments (*p* = 0.747; Table 2). The lowest average plant height was measured in the control treatment (0.40 m), and the highest was for SMC (0.525 m). Plant volume for the SMC treatment was higher than for the control and PAM treatments (41.14 vs. 32.56 and 36.26 L m$^{-1}$, respectively). Notably, no runoff was generated following a cumulative rainfall depth of 142.8 mm in the SMC plots, while the total cumulative rainfall in the control and PAM treatments was similar (*p* = 0.628). Runoff was generated in the PAM treatment following an amount of rainfall almost twofold higher than that generated under the control treatment (25.58 vs. 13.09 mm, respectively). The final difference in RO between the control and PAM treatments was negligible. Similar results were measured for the mean and final RRs of the PAM and control treatments (Table 2).

## 4. Discussion

This study used foxtail millet as a model crop for assessing runoff generation over the growing season. The autumn rainfall simulation, conducted over bare soil, showed that SMC (which provided 35% soil coverage) allows for considerable mitigation of runoff generation compared to the control and PAM treatments. This reduction is probably related to the decrease in raindrop splash impact over the mulch cover compared to the bare plowed surface. These findings accord with previous studies [28,34] that highlighted the benefits of utilizing mulch as a buffer layer to reduce the initiation of runoff and erosional processes. The reduction of raindrop impact on the bare surface negates aggregate slaking and clay dispersion, minimizing mechanical crust formation and lessening runoff generation. The results indicate that the SMC treatment is more effective than PAM at regulating mean and final RRs under similar rainfall conditions. The lack of difference between PAM and control treatments, as revealed in this study, contradicts previous studies conducted under laboratory conditions [49,50]. This suggests that the prevailing environmental conditions play a central role in controlling runoff properties.

The winter rainfall simulation, where lower rainfall intensities were applied over a moderate vegetation cover, did not result in significant differences among the treatments. This demonstrates the raindrop impact-buffering effect of vegetation cover (or mulch), reducing mean and final ROs and RRs. Despite the similarities among the treatments, the higher RO and RR values measured for the PAM compared to the SMC and control treatments demonstrate the inferior effect of the PAM on surface sealing in this field experiment. These results fit those of Jordán et al. [68], who found that adding mulch at a rate of 500.0 kg ha$^{-1}$ delayed runoff generation and soil loss compared to bare soil conditions. Moreover, despite the substantially lower plant height and volume in the SMC treatment, the impact of the lower foliage density on runoff properties was negligible. This consists with Schindler Wildhaber et al. [69], who assessed the impact of vegetation cover on runoff generation under a simulated rainfall of 65 mm h$^{-1}$, and reported no effect on surface runoff.

The spring rainfall-simulation scenario revealed the most noticeable difference between SMC and the other two treatments, where no runoff was measured for the SMC plots following 240 min of rainfall (Table 2). Though vegetation cover on the SMC plots was similar to that on the PAM and control plots, plant height and volume on the SMC plot were higher, suggesting that foliage cover and density enhanced the buffer effect, which further decreased raindrop impact over the mulch cover, thus improving the absorbance of rainfall throughout the entire simulated rainstorm. A comparison between PAM and the control revealed similar values, suggesting that PAM is ineffective as a soil stabilizer in these clayey environments.

## 5. Conclusions

We investigated the effects of PAM and SMC on runoff properties under three rainstorm scenarios in a foxtail millet cropping system. The main insights of this study, achieved using a portable rainfall simulator in the field, are as follows:

- Under bare soil conditions (autumn simulation), the SMC treatment (providing 35% coverage) was better at controlling runoff generation than the PAM treatment. The winter simulation results were inconclusive for all treatments. For the spring simulation, no runoff was recorded from the SMC, whereas runoff from the PAM was similar to that from the control. This implies that the effect of the PAM application diminishes over the growing season and that its effectiveness in runoff control is negligible.
- Following tillering and flowering, vegetation cover did not impact runoff properties. This suggests that the mulch cover protects the soil surface from raindrop impact.
- PAM has little to no effect as a stabilizer in cultivated vertisols, contradicting previous laboratory studies but consistent with other field studies performed in similar environments.

**Author Contributions:** Conceptualization, E.A.; methodology, E.A.; validation, E.A., and I.S.; formal analysis, E.A.; investigation, E.A.; resources, E.A.; data curation, E.A.; writing—original draft preparation, E.A.; writing—review and editing, E.A. and I.S.; visualization, E.A.; supervision, E.A.; project administration, E.A.; funding acquisition, E.A. All authors have read and agreed to the published version of the manuscript.

**Funding:** This research was funded with the support of the Ministry of Agriculture and Rural Development.

**Data Availability Statement:** The data supporting the findings of this study are available from the corresponding author, Eli Argaman, upon request.

**Acknowledgments:** The authors thank Efraim Fizik (of blessed memory), Moshe Gottesman, Alon Maor, and Idit Tokotzki for their skilled technical assistance.

**Conflicts of Interest:** The authors declare no conflict of interest. The funders had no role in the study's design; in the collection, analysis, or interpretation of data; in the writing of the manuscript; or in the decision to publish the results.

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
