# Peer review of "Runoff Mitigation in Croplands: Evaluating the Benefits of Straw Mulching and Polyacrylamide Techniques"

_agronomy, doi:10.3390/agronomy13071935_

Round 1

Reviewer 1 Report

Soil erosion by water is one of the most destructive phenomenon resulting in land degradation. Effect of mulching on runoff generation is a well known fact and it is being used worldwide in many regions however as pointed out by authors the availability of straw mulch is a major issue. The results also indicate that PAM is not effective for runoff control so there is need to try for alternate methods. The sediment yield information is equally important which is lacking in this manuscript. In addition to SMC and PAM other sources or techniques should also be tried and compared with these. Also the study which is done in micro-plots using simulated rainfall is quite similar to laboratory studies. So there is need to go for big plot size to replicate field conditions specially when treatments like mulching are imposed. 

Author Response

Thanks for the provided review that helps to improve this manuscript.

Our results indicate that applying PAM as a stabilizer of clayey soils is inferior to straw mulch cover. Further, the research findings suggest that despite its comparatively high cost, straw mulch cover may be considered a sustainable practice that effectively controls water loss through runoff from agricultural fields. The research field experimental setup was conducted under similar conditions done by others, as described in the introduction.

As described in the manuscript, we repeated each treatment at each simulation three times to improve the research reliability. While most rainfall simulation experiments were conducted in a controlled laboratory environment, the current study was carried out under field conditions and lasted during the cropping season. We agree with the reviewer's comment that a slope scale experiment is essential and might highlight novel insights that were not achievable under the current study. Yet, this study focused on the raindrop-runoff interaction during a long-term field simulation encompassing the whole crop growing phases. The authors are not aware of previous studies that simulated these processes.

Reviewer 2 Report

The paper presents valuable insights into the effectiveness of SMC and PAM treatments for controlling runoff under different rainstorm scenarios. The study design, methodology, and data analysis are sound, and the findings contribute to the existing knowledge in the field of erosion control and water conservation. However in some parts of the paper adjustments and deeper explanations and analysis are needed:

L26-81 a) The literature review / Introduction should clearly state the objective of the research study. It should specify what the study aims to investigate or achieve.

b) Expand on the significance of the issue of water loss through runoff in drylands and its implications for agriculture. Explain why it is important to mitigate runoff and improve water-use efficiency in rainfed agricultural lands across the Mediterranean region.

c) When discussing soil-conservation measures, such as straw-mulch cover and polyacrylamide, provide more specific details about their mechanisms of action, applications, and previous research findings. This will enhance the reader's understanding of the topic. Address the limitations and challenges associated with the use of straw-mulch cover and polyacrylamide.

d) Discuss any conflicting findings or factors that may affect their effectiveness in different contexts.

e) Explicitly state the hypothesis of the study and provide a rationale for why the efficiency of straw-mulch cover is expected to be greater than that of polyacrylamide in controlling runoff generation and reducing surface water loss in clayey soils.

L 82-170

a) Provide a rationale for selecting the Harod Valley: Explain why the Harod Valley was chosen as the study location. Discuss its relevance to the research topic and how it represents the dry sub-humid Mediterranean region.

b) Specify the rainfall simulation device: Provide more information about the portable rotating-disk rainfall simulator used in the experiments. Describe its specifications, working principles, and how it was calibrated to ensure consistent rainfall intensities.

c) Clarify the experimental timeline: Clearly state the duration and timing of the experiments. Specify the start and end dates of each seasonal and phenological phase (autumn, winter, and spring) to give a clear understanding of the study's timeline.

d) Describe plant selection and growth: Provide more details about the selection of foxtail millet (Setaria italica) as the chosen crop for the plots. Explain why this crop was selected, its growth characteristics, and how it relates to the research objectives.

e) Explain plant water management: Provide information on how plant water stress was avoided during the experiment. Describe the irrigation practices used to ensure consistent water availability for all experimental treatments.

f) Clarify runoff data collection: Explain in more detail how runoff data were collected. Provide information on the measurement frequency, techniques used to measure runoff rates, and how the runoff was captured and stored for analysis.

L151 Clarify the interpretation of results: Provide clearer interpretations of the results. Instead of simply stating that certain parameters were "significantly lower" or "similar," explain the implications of these differences or similarities. Discuss the practical significance and potential implications of the observed variations.

L 141 - 201 Report sample sizes: Specify the number of replicates or observations for each treatment and simulation phase. This information is important for understanding the reliability and robustness of the results.

L202-234

a) Add the explanation of the mechanisms: Elaborate on the mechanisms underlying the observed effects of SMC and PAM treatments on runoff generation. Discuss the physical processes, such as raindrop impact, soil sealing, and erosion, and how the treatments influence these processes. Support your explanations with references to relevant studies.

b) Discuss the practical implications and potential applications of your findings. Explain how the reduction in runoff generation achieved with SMC treatment can contribute to erosion control and water conservation efforts. Consider the environmental and agricultural benefits of using SMC as a runoff mitigation strategy.

L 235 - 251 a) Discuss the limited effectiveness of PAM, Reinforce the superiority of SMC treatment, Highlight the role of mulch cover, Address the seasonal variations, Consider the broader implications, Suggest future research directions

Author Response

Thank you for the comprehensive review that improved the manuscript quality.

Please see attached file with our reply

Round 2

Reviewer 1 Report

Dear authors,

The revised manuscript gives better presentation of data and now most of the queries are answered. However, it shall be more beneficial if data on sediment yield is also added if available. 

Regards

Author Response

Thank you for the effort and support to improve our submitted MS.

Please view our reply attached.

Reviewer 2 Report

I am delighted to inform you that I have reviewed the revised version of your manuscript and I am satisfied with the outcome. The revisions made to the manuscript have strengthened its overall quality. The clarity of your writing has improved, making the content more accessible and engaging for readers. In conclusion, I am pleased to inform you that your manuscript has successfully addressed all the concerns raised during the review process, resulting in a good-quality contribution to the field.

Author Response

Thank you for the support, valuable comments, and insights that improved the submitted MS.